# Determination of Antimicrobial Resistance and the Impact of Imipenem + Cilastatin Synergy with Tetracycline in *Pseudomonas aeruginosa* Isolates from Sepsis

**DOI:** 10.3390/microorganisms11112687

**Published:** 2023-11-02

**Authors:** Telma de Sousa, Catarina Silva, Olimpia Alves, Eliana Costa, Gilberto Igrejas, Patricia Poeta, Michel Hébraud

**Affiliations:** 1MicroART-Antibiotic Resistance Team, Department of Veterinary Sciences, University of Trás-os-Montes and Alto Douro, 5000-801 Vila Real, Portugal; telmas@utad.pt (T.d.S.); xusilva2002@gmail.com (C.S.); ppoeta@utad.pt (P.P.); 2Department of Genetics and Biotechnology, University of Trás-os-Montes and Alto Douro, 5000-801 Vila Real, Portugal; gigrejas@utad.pt; 3Functional Genomics and Proteomics Unit, University of Trás-os-Montes and Alto Douro, 5000-801 Vila Real, Portugal; 4Associated Laboratory for Green Chemistry, University NOVA of Lisbon, 1099-085 Caparica, Portugal; 5Hospital Centre of Trás-os-Montes and Alto Douro, Clinical Pathology Department, 5000-801 Vila Real, Portugal; orvalves@chtmad.min-saude.pt (O.A.); ecsvalente@chtmad.min-saude.pt (E.C.); 6CECAV—Veterinary and Animal Research Centre, University of Trás-os-Montes and Alto Douro, 5000-801 Vila Real, Portugal; 7Veterinary and Animal Research Centre, Associate Laboratory for Animal and Veterinary Science (AL4AnimalS), 5000-801 Vila Real, Portugal; 8INRAE, Université Clermont Auvergne, UMR Microbiologie Environnement Digestif Santé (MEDiS), 63122 Saint-Genès-Champanelle, France

**Keywords:** *Pseudomonas aeruginosa*, imipenem + cilastatin, tetracycline, sepsis, antibiotic resistance

## Abstract

*Pseudomonas aeruginosa* is among the most ubiquitous bacteria in the natural world, exhibiting metabolic and physiological versatility, which makes it highly adaptable. Imipenem + cilastatin and tetracycline are antibiotic combinations commonly used to treat infections caused by *P. aeruginosa*, including serious infections such as sepsis. In the context of bacterial infections, biofilm, formed by bacterial cells surrounded by extracellular substances forming a matrix, plays a pivotal role in the resistance of *P. aeruginosa* to antibiotics. This study aimed to characterize a representative panel of *P. aeruginosa* isolates from septicemias, assessing their susceptibility to various antibiotics, specifically, imipenem + cilastatin and tetracycline, and the impact of these treatments on biofilm formation. Results from antibiotic susceptibility tests revealed sensitivity in most isolates to six antibiotics, with four showing near or equal to 100% sensitivity. However, resistance was observed in some antibiotics, albeit at minimal levels. Notably, tetracycline showed a 100% resistance phenotype, while imipenem + cilastatin predominantly displayed an intermediate phenotype (85.72%), with some resistance (38.1%). Microdilution susceptibility testing identified effective combinations against different isolates. Regarding biofilm formation, *P. aeruginosa* demonstrated the ability to produce biofilms. The staining of microtiter plates confirmed that specific concentrations of imipenem + cilastatin and tetracycline could inhibit biofilm production. A significant proportion of isolates exhibited resistance to aminoglycoside antibiotics because of the presence of modifying genes (*aac*(3)-*II* and *aac*(3)-*III*), reducing their effectiveness. This study also explored various resistance genes, unveiling diverse resistance mechanisms among *P. aeruginosa* isolates. Several virulence genes were detected, including the las quorum-sensing system genes (*las*I and *las*R) in a significant proportion of isolates, contributing to virulence factor activation. However, genes related to the type IV pili (T4P) system (*pil*B and *pil*A) were found in limited isolates. In conclusion, this comprehensive study sheds light on the intricate dynamics of *P. aeruginosa*, a remarkably adaptable bacterium with a widespread presence in the natural world. Our findings provide valuable insights into the ongoing battle against *P. aeruginosa* infections, highlighting the need for tailored antibiotic therapies and innovative approaches to combat biofilm-related resistance.

## 1. Introduction

*P. aeruginosa* is a bacillus-shaped pathogen with both aerobic and anaerobic metabolic capabilities. Although primarily known as a human pathogen, this bacterium also has the ability to infect animals and plants [1]. *P. aeruginosa* possesses numerous virulence factors that enable it to damage host cells and manipulate immune defense mechanisms, resulting in a high potential for antibiotic metabolism. Infections caused by this species can lead to severe and occasionally fatal consequences, as this pathogen is progressively becoming more resistant to commonly used antibiotics. Consequently, it is one of the leading causes of therapeutic failures. The continuous administration of antibiotics, accompanied by the occurrence of cross-resistance between multiple pathogens, has contributed to the emergence and prevalence of multidrug-resistant *Pseudomonas aeruginosa* (MDR) strains. These particular strains have exhibited resilience against two or more distinct types of antimicrobial agents [2].

The most common nosocomial infections caused by *P. aeruginosa* are respiratory, gastrointestinal, and urinary tract infections, as well as bloodstream infections [3]. The latter is alarming since blood is a fundamental fluid for transporting molecules to all the cells of the body and can deliver favorable substances but also harmful substances to cells [4,5]. When *P. aeruginosa* enters the bloodstream, it can result in severe complications such as sepsis. Biofilms provide protection to bacteria against the immune system and antibiotics, making these infections particularly challenging to treat [6].

Clinical strains of *P. aeruginosa* often exhibit extensive intrinsic resistance to a wide range of antimicrobial agents, including tetracyclines and β-lactams, making antibiotic therapy a challenge in clinical treatments. Intrinsic resistance has generally been attributed to low membrane permeability, but it can also resist antibiotics that attack the outer membrane through the efficient implantation of transmembrane efflux pumps, preventing contact between antibiotics and their intracellular targets [7]. Imipenem is a broad-spectrum antibiotic that inhibits the synthesis of bacterial cell walls, for example, in Gram-positive bacteria. Multidrug-resistant bacteria are commonly treated with imipenem in order to treat severe infections. Cilastatin is usually used in combination with other antibiotics, namely, imipenem, to prevent it from being degraded by the dehydropeptidase enzyme in the kidneys, helping to keep imipenem levels in the body high. The combination of imipenem and cilastatin is usually used to treat serious infections, especially those caused by multidrug-resistant bacteria, or in patients with impaired kidney function. It should be noted that this drug is generally administered intravenously and requires careful monitoring of kidney function and possible side effects [8].

*P. aeruginosa* is one of the most important model organisms for the study of biofilms because of its relative ease of growth, its consistent structure and reproducibility under laboratory conditions, and its relevance as an opportunistic pathogen in hospital environments [9]. Given the characteristics of biofilms, they are responsible for increasing the resistance of bacteria to antibiotics. This fact is demonstrated by the low penetration capacity of the antibiotic through the extracellular polysaccharide matrix [10].

The purposes of this study were to characterize antimicrobial resistance phenotypes, to determine the minimum inhibitory concentration of imipenem + cilastatin and tetracycline, to investigate biofilm development with and without imipenem + cilastatin and the synergistic effect of these antibiotics, and to draw up an assessment of antibiotic resistance and virulence genes in this panel of *P. aeruginosa* isolates from septicemia cases.

## 2. Materials and Methods

### 2.1. Bacterial Strains and Culture Media

In this study, twenty-one samples of septicemia of human origin from the Centro Hospitalar de Trás-os-Montes and Alto Douro (CHTMAD) were studied. All strains were isolated using VITEK 2^®^ COMPACT (bioMérieux, Minato, Tokyo, Japan). After collecting the samples, their identification was confirmed by removing a colony with a sterilized loop and spreading it on a Petri dish containing *Pseudomonas* agar base medium with CN supplement (Liofilchem^®^ s.r.l., Roseto degli Abruzzi, Italy), a selective medium for *P. aeruginosa* growth. Subsequently, Petri dishes were incubated for 24–48 h at 37 °C, and, after confirmation, the isolates were cryopreserved at −20 °C in skim milk.

### 2.2. Kirby–Bauer Disc Diffusion Assay

Bacterial cell suspensions in saline (0.9%) were normalized at 0.5 McFarland standard and swabbed onto Mueller Hinton (MH) (Frilabo, Maia, Portugal) agar plates. Disks containing amikacin (AK, 30 μg), cefepime (FEP, 30 μg), ceftazidime (CAZ, 30 μg), ciprofloxacin (CIP, 5 μg), doripenem (DOR, 10 μg), gentamicin (CN, 10 μg), imipenem (IMI, 10 μg), meropenem (MEM, 10 μg), piperacillin (PRL, 30 μg), ticarcillin–clavulanic acid (TTC, 85 μg), and tobramycin (TOB, 10 μg) were placed on the surface of inoculated plates, and diameters of growth inhibition zones were measured after 24 h of incubation at 37 °C. These antibiotics and their concentrations followed the EUCAST 2022 standards [11], except for the antibiotic ceftazidime, which had CLSI 2021 as a reference [12].

### 2.3. MIC Assays and Synergy between Imipenem + Cilastatin and Tetracycline

The MIC (minimum inhibitory concentration) levels of imipenem + cilastatin (500 mg + 500 mg) and tetracycline were determined via the broth microdilution method in MH II broth (Frilabo, Portugal). Strains were grown at 37 °C for 24 h at 150 rpm and then diluted at 5 × 10^5^ cells/mL in the presence of increasing concentrations of imipenem + cilastatin (up to 0–16,384 mg/L) and tetracycline (up to 128 mg/L). Bacterial growth was determined at 490 nm using a microplate reader (Biochrom, EZ Read 800Plus, London, UK). The MIC was recorded after 24 h at 37 °C at 150 rpm. Each strain was tested in at least eight independent experiments.

The synergy between the antibiotics was determined by measuring the fractional inhibitory concentration (FIC) using a checkerboard synergy assay. Bacterial strains were grown in MH II broth and then diluted at a concentration of 5 × 10^5^ cells/mL in this medium. Each bacterial suspension was placed across the entire 96-well plate. Meanwhile, the tetracycline antibiotic, corresponding to the columns, was placed in a concentration range between 0 and 256 mg/L. In contrast, imipenem + cilastatin corresponded to the rows on the plate and was also placed in the same range of concentrations. Wells without antimicrobial agents but with bacteria acted as positive controls, and wells with only medium acted as negative controls. The degree of synergy was determined by calculating the fractional inhibitory concentration index (FICI). The formula used was as follows: FIC = (MIC of antibiotic A in combination/MIC of antibiotic A alone) + (MIC of antibiotic B in combination/MIC of antibiotic B alone). Synergism was defined when FIC ≤ 0.5 [13].

### 2.4. Evaluation of Biofilm Total Biomass Using Crystal Violet Staining Assay

Biofilm growth was performed with modifications to previous methods from the literature [14]. One colony of each sample of *P. aeruginosa* from the BHI agar (Frilabo, Portugal) medium was transferred to different test tubes with liquid LB (Oxoid^®^, Basingstoke, UK) incubated for 24 h at 150 rpm in an incubate at 37 °C. Then, the cultures were diluted in fresh medium TSB (Oxoid^®^, Basingstoke, UK) until a concentration equivalent to 1.5 × 10^8^ cells/mL was obtained. Bacteria were placed in 96-well microplates. Two columns of each plate served as a positive control (ATCC^®^ 27853) and, as a negative control, with only TSB to rule out possible contamination. After this procedure was carried out in the flow chamber, the plates were placed in the incubator for 24 h at 37 °C and 150 rpm. Staining was performed using 0.1% crystal violet. Finally, 30% acetic acid was added (to “disassociate” the violet crystal from the biofilm), and the absorbances of the plates were read at 630 nm in a plate reader (Biochrom, EZ Read 800 Plus). Subsequently, the following relationships were used to quantify the formed biofilms: OD ≤ ODC (non-producer), ODC < OD ≤ 2ODC (weak), 2ODC < OD ≤ 4ODC (moderate), OD < 4ODC (strong).

### 2.5. Biofilm Formation with Imipenem + Cilastatin and Tetracycline

Biofilms were generated as described in the previous section. Strains were grown at 37 °C for 24 h at 150 rpm and then updated by diluting them at 5 × 10^5^ cells/mL in the presence of increasing concentrations of imipenem + cilastatin (up to 0–16,384 mg/L) and tetracycline (up to 128 mg/L). Plates from synergy results were stained with 0.1% crystal violet followed by 30% acetic acid. In this way, we could verify the presence or absence of biofilms depending on the different concentrations of the two studied antibiotics.

### 2.6. DNA Extraction

The boiling method [15] was used to extract DNA. Briefly, two to three bacterial colonies grown overnight were suspended in a test tube containing 500 µL of distilled water and boiled for 8 min in a water bath. After vigorous vortexing, the samples were centrifuged at 12,000 rpm for 2 min, and the pellet was discarded. The total DNA concentration was determined using the NanoDrop system. A measurement of absorbance was taken at wavelengths of 260 and 280 nm to determine the concentration of nucleic acids. Each optical density unit corresponded to 50 μg/mL of double-stranded DNA. The best purities were 1.8 and 2 for all DNA samples.

### 2.7. Detection of Antibiotic Resistance Genes

The presence of antibiotic-resistance genes was investigated in all isolates via PCR. Several primers were tested for genes encoding resistance to different antibiotics: class A ESBLs (*bla*_TEM_, *bla*_SHV_, *bla*_CTX_, *bla*_PER_), class A carbapenemases (*bla*_KPC_, *bla*_IMI_), class B Metalo-β-lactamases (MBLs) (*bla*_SPM_, *bla*_IMP_, *bla*_Vim_, *bla*_Vim-2_, *bla*_NDM_), class D β-lactamics (*bla*_OXA_, *bla*_OXA-48_), fluoroquinolones (*qnr*S, *qnr*A), aminoglycosides (*aac*(6″)-*aph*(2″), *aac*(3)-II, *aac*(3)-III, aac(3)-IV, *ant*(6′)-Ia, *ant*(4′)-Ia and *ant*(2′)-Ia), and efflux pump (*oprD*). All primer sequences were the same as already published in de Sousa et al. [16].

### 2.8. Detection of Virulence Genes

All isolates were screened for genes encoding virulence factors via PCR: genes encoding type IV pili (T4P) (*pil*B, *pil*A), alkaline protease (*apr*A), exotoxin A (*tox*A), T6SS effector proteins (*tss*C), phospholipase (*plc*H), elastases (*las*A, *las*B), the *las* quorum-sensing system (*las*R, *lasI*), T3SS effector genes (*exo*U, *exo*S, *exo*A, *exo*Y, *exo*T), *rhl* quorum-sensing system genes (*rhl*R, *rhl*I, *rhl*A/B), and alginate (*alg*D). All primer sequences were the same as already published in de Sousa et al. [16].

## 3. Results and Discussion

### 3.1. Determination of Sensitivity to Antibiotics

The isolates showed sensitivity to 6 out of the 11 antibiotics examined (Figure 1). Among these, amikacin and tobramycin demonstrated full susceptibility in all 21 strains (100%). Additionally, 18 out of 21 strains (85.71%) exhibited susceptibility to gentamicin and meropenem, while only 1 of the 21 strains displayed sensitivity to ceftazidime and ciprofloxacin. It is noteworthy that the first three antibiotics mentioned above belong to the aminoglycoside class. Furthermore, we observed a limited number of resistant strains (ranging from 2 to 7) for most of the other antibiotics under investigation. These findings were consistent with those reported in the study by Anjum and Mir, where effective values were also obtained for amikacin and tobramycin (with sensitivities of 79% and 70%, respectively) [17]. This status indicates that these bacterial strains exhibit an intermediate response, signifying that they are less sensitive to the antibiotic compared with susceptible strains but have not yet reached a state of full resistance. The clinical significance of these intermediate results varies depending on several factors, including the specific antibiotic under consideration, the type of infection being treated, and the overall condition of the patient. In some cases, intermediate results may still allow for successful treatment, while in others, alternative antibiotics may be preferred.

Regarding cephalosporin antibiotics such as cefepime and ceftazidime, the isolates generally showed a predominantly intermediate phenotype. However, a study conducted by Anjum et al. reported a high percentage of ceftazidime-sensitive isolates, specifically, 62% [17]. An investigation carried out in Egypt showed that 51% of isolates were cefepime-resistant [18].

It was observed that meropenem and imipenem, both belonging to the same class of antibiotics, demonstrated markedly divergent efficacy against *P. aeruginosa*. The imipenem exhibited greater effectiveness compared with meropenem [19]. However, in the specific context of this study, the range of efficacy values for imipenem was notably wider than previously documented in similar studies. The variance in efficacy between meropenem and imipenem against *P. aeruginosa* could be attributed to several factors. Despite both antibiotics falling within the same carbapenem class, subtle distinctions in their chemical structures and pharmacological characteristics might account for variations in their antibacterial activity [20]. In addition, factors such as the administered dosage, frequency of administration, and duration of treatment can also influence the effectiveness of antibiotics [21]. It is possible that, in this study, the most appropriate dosage, administration scheme, or duration of treatment was used for imipenem, resulting in greater efficacy compared with meropenem.

Regarding ciprofloxacin, our study indicated a lower percentage of sensitive isolates, which contrasted with findings from other studies. It should be noted that the results obtained by Lister [19] differ from those of other studies [17,18]. These discrepancies can be attributed to differences in the populations of *P. aeruginosa* studied, the methodologies used in antibiotic sensitivity tests, and other factors that may influence bacterial resistance like the fact that these studies are over 10 years old and that the profiles and numbers of resistant strains have changed since then.

### 3.2. MIC Assays and Synergy between Imipenem + Cilastatin and Tetracycline

*P. aeruginosa* had a 100% resistance to the antibiotic tetracycline, with the majority of strains having an MIC value of 32 or 64 mg/L (Figure 2). More specifically, the MIC of 64 mg/L, observed in twelve isolates (57.15%), was what appeared to be the most predominant among the various isolates. For imipenem + cilastatin, the phenotype was generally intermediate (85.72%), although it also presented some resistance (14.28%). Additionally, eight isolates (38.1%) displayed an intermediate phenotype with an MIC of 4 mg/L.

Similar to the results associated with tetracycline, a study also revealed MIC results equal to 32 mg/L [22], while another study also showed MIC values equal to or greater than 8 mg/L, which goes against the results shown here [7]. A study carried out in Egypt by Gad [18] stated that most isolates had an inhibitory concentration of 64 mg/L. This study also obtained high values of inhibited isolates with concentrations of 16 and 32 mg/L. The consistency of resistant phenotypes (MIC > 8 mg/L) suggests that *P. aeruginosa* employs highly effective mechanisms against this antibiotic. According to Li [23], while resistance to tetracycline is due to the low permeability of the outer membrane, this is not the only reason for it. The notable resistance levels and the diversity of results suggest the involvement of an additional factor. This factor may relate to mechanisms that either break down or alter antibiotic molecules, essentially leading to antibiotic removal, implying a potential role of efflux pumps in conferring resistance [23].

In the case of the antibiotic imipenem + cilastatin, the results obtained by Ranji and Rahbar Takrami differed from our results [24]. This study consisted of observing the resistance of 22 different strains of *P. aeruginosa* to the antibiotic imipenem. The exposed results essentially revealed a resistant phenotype with MICs equal to or greater than 8 mg/L, except for two situations in which bacterial growth inhibition was achieved at 2 and 4 mg/L [24]. Our results revealed an essentially intermediate phenotype and lower MICs. Therefore, from the comparison of these two studies, we confirm that cilastatin seems to have a fundamental role in inhibiting bacterial growth since a lower concentration of antibiotics is required. Imipenem is an antibiotic of the carbapenem group, which has activity against a wide variety of Gram-positive and Gram-negative bacteria. This acts by inhibiting the synthesis of the bacterial cell wall, leading to bacterial cell death [8]. Cilastatin, in turn, is an inhibitor of the dehydropeptidase I enzyme. This enzyme normally degrades imipenem in the renal tract, resulting in a significant reduction in its concentration and, therefore, its effectiveness. By inhibiting dehydropeptidase I, cilastatin prevents the degradation of imipenem, thus increasing its effective concentration in the body [25]. Therefore, imipenem and cilastatin are two compounds often combined into a single imipenem + cilastatin product that allows imipenem to be administered effectively without being rapidly eliminated by the renal tract. This combination improves its concentration and time of action in the body, increasing its effectiveness in the treatment of severe bacterial infections [25], which most likely explains the greater influence of this antibiotic in our study. A comparison of the effectiveness of imipenem + cilastatin and tetracycline antibiotics can be carried out based on their mechanism of action and other types of factors, such as frequency of administration. Carbapenems are able to resist the hydrolytic action of the β-lactamase enzyme, which makes them more effective in fighting bacterial infections [26]. More specifically, imipenem achieves a broad spectrum of efficacy against both aerobic and anaerobic pathogens and is, therefore, normally administered orally and in low concentrations [27]. On the other hand, tetracycline is an antibiotic that had high efficacy in the past. However, given its constant use and, consequently, the growing bacterial resistance to this class of antibiotics, it is no longer as efficient [28].

In our synergy test, 6 of the 21 isolates were selected from this study. These samples were chosen because they had MIC values for both tetracycline and imipenem + cilastatin that were below or above average. Various combinations have proved effective against different isolates. These combinations include 16 mg/L tetracycline and 2 mg/L imipenem + cilastatin; 32 mg/L tetracycline and 2 mg/L imipenem + cilastatin; 8 mg/L tetracycline and 4 mg/L imipenem + cilastatin; 16 mg/L of tetracycline and 16 mg/L of imipenem + cilastatin; and 8 mg/L of tetracycline and 64 mg/L of imipenem + cilastatin. In addition to these synergistic values, many other concentrations of the two antibiotics have been found to be additive: 4 ≥ FIC > 0.5 [13].

In a study that analyzed the synergy of tetracycline with several antibiotics, a synergistic character was specifically observed with amoxicillin/clavulanic acid, an antibiotic of the β-lactam family, the same as imipenem [13]. Tetracycline is a bacteriostatic antibiotic that causes the inhibition of protein synthesis by binding to the 30S ribosomal subunit. This antibiotic, combined with β-lactam antibiotics, reveals synergism. This may be a result of the increased permeability caused by β-lactams and the consequent penetration of tetracycline into the cell and subsequent inhibition of protein synthesis [29,30]. Thus, several studies prove the synergistic interaction between β-lactams and other antibacterial agents, in which the inhibition of cell wall synthesis caused by β-lactams significantly enhances the absorption of other drugs [31]. Thus, there is a decrease in the necessary concentration of each antibiotic, even though the necessary concentration of tetracycline is higher in the vast majority of cases since its initial MIC was also higher compared with the MIC of imipenem + cilastatin.

### 3.3. Evaluation of Total Biofilm Biomass with and without Imipenem + Cilastatin and Tetracycline

Overall, the results obtained showed that our panel of 21 *P. aeruginosa* isolates from human septicemia was capable of forming mainly moderate biofilms. A single isolate demonstrated a robust ability to develop strong biofilms. None of the strains proved incapable of forming biofilms or even weak biofilms, confirming this well-known characteristic of the *P. aeruginosa* species [32,33], regarding the influence of temperature on biofilm formation, showed that some have studies highlighted the production of biofilms by *P. aeruginosa* at 37 °C, which is the temperature commonly found in the human body and the optimal growth temperature for *P. aeruginosa* [33,34]. Temperature can influence the expression of genes related to biofilm production, as well as the formation and structure of biofilms [35]. The optimum temperature for biofilm formation can vary between different strains of *P. aeruginosa* and can also be affected by other environmental factors, nutrient availability, pH levels, or oxygen concentrations. The effects of representative environmental (23 °C) and host (37 °C) temperatures on biofilms formed by *P. aeruginosa* were analyzed by Bisht et al. [36]. They found that a temperature of 37 °C favored the formation of denser biofilms with a more complex architecture compared with 23 °C, in which the biofilm matrix seemed to be slightly disseminated. Another study investigating the influence of temperature on the formation of biofilms caused by *P. aeruginosa* found that different proteins were expressed by sessile cells according to the temperatures, revealing the adaptation of this bacterial species to different conditions. However, the most robust and structurally complex biofilms were produced [34], such as those formed in our assays. A study covering a wider range of temperatures (28 °C, 33 °C, 37 °C, and 42 °C) also revealed that 37 °C was the temperature at which biofilms were formed with greater mechanical stability [37]. From this study, we can also add that, possibly, the fever immune response can help in the fight against infections of this bacterial type. These studies highlighted that temperature, specifically, 37 °C, can provide ideal conditions for the formation of denser and more complex biofilms caused by *P. aeruginosa*. This is particularly relevant, as the human body temperature is around 37 °C, which suggests that *P. aeruginosa* takes advantage of physiological conditions to increase the metabolic rate of its cells, consequently increasing the rate of production of extracellular material [37]. Thus, human body temperature seems to be a favorable condition for this bacterium, which is why infections caused by this bacterium are particularly challenging.

The six isolates (HS2, HS7, HS12, HS17, HS18, HS20) used for our synergistic tests were also stained with 0.1% crystal violet to verify whether the two antibiotics in combination have any influence on biofilm formation (Table 1). It clearly appeared that the combination of imipenem + cilastatin and tetracycline was capable of inhibiting the production of biofilms but that the effective concentrations depended on the isolates. For example, a concentration equal to or greater than only 2 mg/L of imipenem + cilastatin but with 256 mg/L of tetracycline was sufficient to prevent biofilm production by the HS2 isolate. In contrast, this effect was observed for isolate HS7 with only 2 mg/L of tetracycline but 16 mg/L of imipenem + cilastatin. Thus, these results demonstrated a dose/response relationship between imipenem + cilastatin and tetracycline concentrations. This means that, when certain concentrations are reached, there is a significant reduction in or complete absence of bacterial biofilm formation. In concrete terms, if the concentration of imipenem + cilastatin is increased to 4 mg/L, the concentration of tetracycline needed to obtain the same biofilm inhibition effect decreases to 128 mg/L (relative to sample HS2). These inhibitory effects of imipenem + cilastatin and tetracycline combinations on bacterial biofilm formation are supported by other studies [38,39]. Furthermore, this suggests that the inhibitory effect can be potentiated or achieved with lower concentrations of one antibiotic when combined with higher concentrations of the other.

Liaqat et al. found that tetracycline hinders protein synthesis, thereby impacting biofilm formation, although its effect is relatively modest [39]. This aligns with the results in Table 2, which indicate the need for a higher antibiotic concentration. In contrast, Piri-Gharaghie et al. suggest that combining imipenem with other antibiotics can increase strain resistance levels [38].

It is important to note that our results regarding the inhibition of biofilm production were specific to the concentrations of imipenem + cilastatin and tetracycline that were tested. Consequently, the interpretation and application of these results must be made with caution, taking into account the specific conditions of the bacterial infection in question, and one must consider the safe and effective doses recommended for the clinical use of antibiotics. This study also demonstrated that the antibiotic concentrations required to effectively inhibit biofilm formation were higher than the minimum inhibitory concentration values. This finding emphasizes that inhibiting bacterial growth does not necessarily guarantee the eradication of the biofilm.

### 3.4. Genotypic Characterization

#### 3.4.1. Detection of Antibiotic-Resistant Genes

The genotypic results for the rDNA 16S gene were positive for all isolates, confirming their identification as *P. aeruginosa*, as expected. Although an analysis was carried out on multiple genes, within the class of aminoglycosides, 15.79% of the isolates had the *aac*(3)-II gene, and 10.53% had the *aac*(3)-III gene (Table 2). These genes encode enzymes called acetyltransferases that have the ability to chemically modify aminoglycoside antibiotics, like gentamicin and amikacin. This modification prevents the antibiotics from binding to their bacterial target, thus reducing their effectiveness in fighting infections. By acetylating aminoglycosides, the enzymes encoded by the *aac*(3)-II and *aac*(3)-III genes change the electrical charge of antibiotic molecules, making it difficult for them to enter bacteria or preventing their binding to bacterial ribosomes, where the synthesis of proteins occurs. As a result, bacteria that have these resistance genes can resist the effects of aminoglycosides and continue to multiply even in the presence of these antibiotics [40].

One study aimed to investigate the diversity and distribution of resistance markers, with a specific focus on the *aac*(3)-II and *aac*(3)-III genes in high-risk international clones of *P. aeruginosa*. High-risk clones are known for their ability to spread and cause difficult-to-treat infections, often associated with multidrug resistance [41]. Molecular techniques, such as polymerase chain reaction (PCR), were used to detect the presence of the *aac*(3)-II and *aac*(3)-III genes in the isolates. The results revealed a diverse distribution of resistance markers among international high-risk clones of *P. aeruginosa*. The *aac*(3)-II and *aac*(3)-III genes, which confer resistance to aminoglycosides, were identified in a significant proportion of the isolates. Interestingly, variations in the presence and prevalence of these genes were observed between different high-risk clones and geographic regions. Furthermore, the study investigated the genetic context of the *aac*(3)-II and *aac*(3)-III genes, analyzing their association with mobile genetic elements, integrons, and other genetic determinants of resistance. This analysis provided information on the mechanisms underlying the dissemination and acquisition of these resistance genes in the Pseudomonas aeruginosa population. The results highlight the importance of monitoring the presence and dissemination of resistance markers, such as the *aac*(3)-II and *aac*(3)-III genes, in high-risk clones of *P. aeruginosa.* Understanding the diversity and distribution of these markers can contribute to the development of targeted surveillance strategies and the implementation of effective infection control measures to mitigate the impact of multidrug-resistant *P. aeruginosa* infections in healthcare settings. Another study aimed to evaluate the rates of aminoglycoside resistance among *P. aeruginosa* isolates obtained in French general practice settings, including community and private health centers. The results revealed that 7% of the isolates had the *aac*(3)-II gene, and 5.8% had *aac(*3)-I. Further analysis of resistance mechanisms indicated that the *aac(*3)-II and *aac*(3)-III genes were responsible for encoding aminoglycoside-modifying enzymes, which chemically modify the structure of aminoglycoside antibiotics and render them ineffective against bacteria [42].

For the identification of genes encoding class D β-lactamase enzymes, 42.11% of the isolates were shown to have the *bla*_OXA-48_ gene in their genome, and no *bla*_OXA_ gene was detected. The *bla*_OXA-48_ gene is associated with resistance to β-lactam antibiotics, specifically in *P. aeruginosa.* The presence of the blaoxa-48 gene in strains of *P. aeruginosa* limits the effectiveness of β-lactam antibiotics, as the enzyme produced by this gene can rapidly degrade and render these antibiotics ineffective. This can lead to treatment failures and challenges in managing infections caused by these resistant strains [43]. Our study is supported by results obtained by a study by Tarafdar et al., which indicated high rates of antimicrobial resistance among strains of *P. aeruginosa* and *Acinetobacter baumannii* isolated from burn infections. Importantly, the study found a significant percentage prevalence of the *bla*_OXA-48_ gene, 70.83% in *P. aeruginosa* and 92% in *A. baumannii*, indicating its involvement in resistance to β-lactam antibiotics [43]

Regarding the detection of class B MBLs, multiple genes were identified. The percentages for the different genes were 68.42% for *bla*_SPM_; 10.53% for *bla*_IMP_; and 5.26% for the *bla*_NDM_, *bla*_PER_, *bla*_Vim_, and *bla*_Vim-2_ genes. Detecting class B MBLs in bloodborne *P. aeruginosa* represents a significant challenge in clinical settings. *P. aeruginosa* is an opportunistic pathogen known for its ability to cause serious infections; MBLs are enzymes produced by certain bacteria that confer resistance to a wide range of β-lactam antibiotics such as penicillins, cephalosporins, and carbapenems [44]. The presence of MBLs in strains of *P. aeruginosa* isolated from blood samples may complicate treatment options and increase the risk of treatment failure. This is of particular concern, as bloodstream infections caused by MBL-producing strains of *P. aeruginosa* are associated with high mortality rates [45]. A study by Marra et al. aimed to investigate the epidemiology, microbiology, and clinical outcomes of bloodstream infections [46]. Among the 76 *P. aeruginosa* isolates analyzed in the study, the MBL *bla*_SPM_ gene was the most common, present in four isolates (5.3%) associated with bloodstream infections. The *bla*_IMP_ gene was detected in two strains (2.6%), while the blaIMP-16 gene was found in one strain (1.3%). No occurrences of the *bla*_Vim_ MBL gene were observed in the isolates under investigation; nevertheless, we should not disregard the low detection of this gene, as its spread can occur quickly, as it is one of the most prevalent emerging types of MBL genes in the world.

Class A carbapenemases are a group of enzymes that can hydrolyze carbapenem antibiotics [44]. In this work, only the *bla*_KPC_ gene was detected in a very small number of isolates, around 5.26%. A study aimed to investigate the occurrence of the *bla*_KPC_ gene in clinical isolates of *P. aeruginosa* collected in several health establishments in Brazil. Of 566 clinical isolates, 86 (15.2%) were found to be positive [47]. A study in Iran also showed the presence of *bla*_KPC_ bacteria in the hospital, which was high [48]. There may be several reasons that justify the low occurrence of the *bla*_KPC_ gene in clinical isolates of *P. aeruginosa* in our study, and in Brazil, in comparison with the fact that high percentages were observed in Iran, it was found that regional variation due to a prevalence of carbapenemase-producing bacteria can vary geographically in terms of timeframe and evolution; however, the study might have been conducted during a period when the prevalence of *bla*_KPC_-positive *P. aeruginosa* isolates was relatively low. Other factors such as geographic differences, local infection control practices, and the specific focus of each study may also interfere with this discrepancy.

Class A extended-spectrum β-lactamases (ESBLs) are enzymes produced by some bacteria that confer resistance to a wide range of β-lactam antibiotics, including cephalosporins and monobactams. ESBLs are a major component of concern in clinical settings because they limit the effectiveness of these commonly used antibiotics, which leads to ESBL infections caused by productive organisms that are more difficult to treat [44]. In this work, only *bla*_TEM_ was detected in 5.26% of isolates. TEM-type ESBLs are widely distributed and have undergone various mutations that confer resistance to extended-spectrum cephalosporins. In this study, only one sequence of the *bla*_TEM_ gene was examined. However, considering that this gene is known to have multiple mutations, it would be worthwhile to screen other sequences to capture the full range of available mutations. A study in Egypt detected *bla*_TEM_ in 23.8% of the isolates [49]. The samples in this study were from camels, but it is extremely important to understand the potential for the transmission of these strains to humans, as it is an animal reservoir, through horizontal gene transfer or even because of the risk of human infection. Another study carried out in Africa detected several ESBLs, like *bla*_TEM_, *bla*_SHV_, and *bla*_CTX-M_, which were detected in 79.3%, 69.5%, and 31.7% of isolates (*n* = 82), respectively [50]. On the other hand, in a study in Iran of 75 ESBL-positive isolates, *bla*_TEM_ (26.7%) was the most common gene, followed by *bla*_CTX-M-15_ (17.3%), *bla*_SHV-1_ (6.7%), and *bla*_SHV-1_ (4%), either alone or in combination [51]. *P. aeruginosa* is a species of bacteria with high genetic diversity, which means that strains may have different levels of resistance, including the presence or absence of ESBL genes; genetic diversity in a strain contributes to observed variations in ESBL detection in assays of various forms. Given the wide genetic diversity of *P. aeruginosa* strains, it is expected that isolates may possess different levels of resistance, including the presence or absence of ESBL genes, so it is important that more sequences are analyzed for comprehensive analysis. Based on the cited analysis, it is warranted to analyze additional sequences to capture the full range of mutations in ESBL genomes [52].

All strains in our study contained the *opr*D gene, which encodes a substrate-specific outer membrane porin in *P. aeruginosa*. This porin facilitates the diffusion of basic amino acids and the small peptide imipenem into the bacterial cell [16]. However, the *opr*D gene shows high polymorphism [53]. This polymorphism may result in the activation of the oprD porin in the presence of insertions that interrupt the *opr*D gene and is directly associated with carbapenem resistance in different strains.

#### 3.4.2. Detection of Virulence Genes

Several virulence genes were detected for the different isolates (Figure 3). For genes that encode the *las* quorum-sensing system, 84.21% and 57.89% were detected for the *lasI* gene and the *lasR* gene, respectively. The *lasI* gene encodes the LasI protein, which is an acyl-homoserine lactone (AHL) synthase, and the *lasR* gene encodes the LasR protein, which is a transcriptional regulator. The *las* quorum-sensing system is known to play an important role in the regulation of virulence in *P. aeruginosa* because it contributes to the infection of this bacterium by activating the expression of genes involved in virulence factor production, such as elastase, pyocyanin, and exotoxin A [54]. The *rhl* quorum-sensing system is another important regulatory network in *P. aeruginosa*, working in conjunction with the las quorum-sensing system. The *las* system is first activated during the early stages of growth, while the *rhl* system is activated later during the transition from the exponential to stationary phase. In our study, the *rhl*R gene and the *rhl*I gene were detected in 100% of isolates. On the other hand, the *rhl*A/B gene appeared in 84.21% of isolates. The high detection rates for the *rhlR* and *rhlI* genes, observed in 100% of the isolates, indicate the widespread presence and activity of the *rhl* quorum-sensing system in the tested *P. aeruginosa* strains. The presence of the *rhlA/B* genes in a high percentage of isolates further supports the involvement of the *rhl* system in the production of virulence factors [55,56].

For the type III secretion system (T3SS), there are several genes that encode different proteins. For such proteins, different *exo* genes have been detected. In this work, *exo*S was detected in all isolates, and all remaining *exo*A, *exo*Y, and *exo*T genes were detected in 78.95% of the isolates. Comparing our results with other studies, it should be noted that these genes are detected with high frequency in *P. aeruginosa* isolates regardless of infection [57,58]. According to another study, out of 132 isolates from various sources, including burn, wound, urine, cystic fibrosis (CF), acute pneumonia, and blood, 120 isolates (91%) were found to secrete the exoS protein based on immunoblot assay results [59]. Additionally, Rumbaugh et al. investigated a collection of 25 isolates obtained from urinary, wound, and tracheal sources and found that 24 isolates (96%) possessed the *exo*S gene [60]. Nevertheless, our results show that not all isolates have the capacity to produce exoS. Our study suggests that the actual overall prevalence of the *exo*S gene in clinical isolates is around 70%. The *exo*U gene was not detected in any isolate. Interestingly, and according to our results, the presence of the *exo*U gene in the literature does not appear consistently and is related to the type of infection.

The *apr*A, *tss*C, and *plc*H genes were detected in all isolates. These genes are responsible for the membrane-bound ATP-binding cassette (ABC), the T6SS system, and phospholipase C, respectively [61,62]. On the other hand, the *alg*D gene was detected in 94.74% of the isolates, and the *tox*A gene was detected in 78.95% of the isolates. These two genes have significance in biofilm formation, adaptation, and cytotoxicity. Studies have demonstrated the high prevalence of *apr*A; for example, Ghanem et al. detected the gene in 85.6% of isolates and the gene *plc*H in 75.2% [63]. The *tss*C gene, despite not directly influencing biofilm formation, is associated with biofilm-specific antibiotic resistance [62]. As anticipated, the presence of this gene was observed in all strains, exhibited biofilm production, and demonstrated varying levels of resistance to antimicrobial agents. Other studies corroborate ours in relation to the high percentage of detection of *tox*A and *alg*D genes [63,64]

*las*B genes were found in all strains, while *las*A genes appeared in 26.32% of isolates. Usually, these genes encode proteins secreted by the type II secretion system, which plays a major role during infection. Numerous studies have examined the occurrence of lasB in *P. aeruginosa* strains sourced from diverse clinical and environmental origins. In one particular investigation, a set of 60 *P. aeruginosa* strains derived from clinical specimens, water sources, and soil were analyzed. Notably, all strains examined were found to possess the *las*B gene [65]. Furthermore, the study identified multiple variants of lasB exhibiting distinctive molecular weights and substrate specificities. This observation implies that *P. aeruginosa* exhibits a diverse repertoire of lasB enzymes, which have likely evolved to accommodate various environments and hosts. The prevalence of *las*B in other studies is also high, which indicates that, in isolates of *P. aeruginosa*, this gene is highly distributed [63,64,66]. Variation in *las*A gene recognition may influence the specific populations and clinical settings studied. Patient populations, clinical samples, and different infections may exhibit different levels of the presence of the *las*A gene. For example, *las*A gene expression or mutation may be low in certain diseases or patient groups [67].

The genes *pilB* and *pil*A encode components of the type IV *pili* (T4P) system in many bacteria, including *P. aeruginosa*. Only one strain expressed the *pil*B gene (5.26%), and some strains had the *pil*A gene (36.84%), which was expected given the low prevalence of this gene; their expression also varies with the infection type [68,69,70,71,72,73]. Both genes are required for type IV *pili* formation and function. These *pili* play important roles in viral infection and interactions with the environment. They facilitate surface adhesion, facilitate bacterial aggregation and biofilm formation, promote bacterial attachment, and promote bacterial motility. A study found that a mutant strain of *P. aeruginosa*, with a deletion in the *pil*B gene, had reduced twitching motility and reduced biofilm formation [73]. On the other hand, multiple studies have reported that *P. aeruginosa* mutant strains lacking the *pil*A gene exhibit significant impairments in biofilm formation on abiotic surfaces and demonstrate reduced virulence in mouse models of infection [72,74]. For blood samples, the *pil*B and *pil*A genes play important roles, enabling *P. aeruginosa* to enter the bloodstream, harbor endothelial cells, and establish systemic infection. Knowledge of the function of these genes in *P. aeruginosa* in blood samples provides valuable insights into the mechanisms underlying bloodstream infections caused by this pathogen. If infections are successfully prevented, this may contribute to targeted interventions focusing on treatment.

## 4. Conclusions

*P. aeruginosa* is a pathogen that represents a significant challenge in the treatment of infections because of its adaptability and diversity of resistance mechanisms. This study highlights the importance of developing new treatment strategies in order to face the multidrug resistance of this bacterial agent. Although *P. aeruginosa* has shown to be 100% sensitive to amikacin, the solution involves the need to combine two antibiotics since the isolated action of only one antibiotic is not as efficient except at very high concentrations, which can be toxic to the individual. With this study, we concluded that the combination of the antibiotic imipenem + cilastatin and tetracycline is beneficial since lower concentrations of each antibiotic are required to inhibit bacterial growth. However, it is essential to understand the differences in the synergistic values obtained, an explanation that will most likely pass through the genetic part of *P. aeruginosa*. Several resistance and virulence genes were detected in different isolates, proving the importance of this study. The next steps involve investigating the mechanisms of synergy and further genetic analysis, like exploring the specific mechanisms underlying the synergistic effect observed between imipenem + cilastatin and tetracycline and delving deeper into the genetic aspects of *P. aeruginosa* to gain a better understanding of variations in synergistic values and resistance mechanisms. This comprehensive approach may require whole-genome sequencing and in-depth genomic studies, potentially leading to the development of more targeted therapies.

## Figures and Tables

**Figure 1 microorganisms-11-02687-f001:**
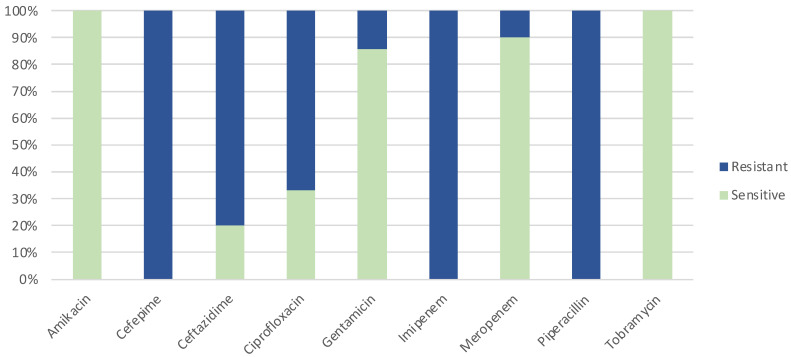
Results of antibiotic sensitivity tests on the panel of *P. aeruginosa* strains (*n* = 21).

**Figure 2 microorganisms-11-02687-f002:**
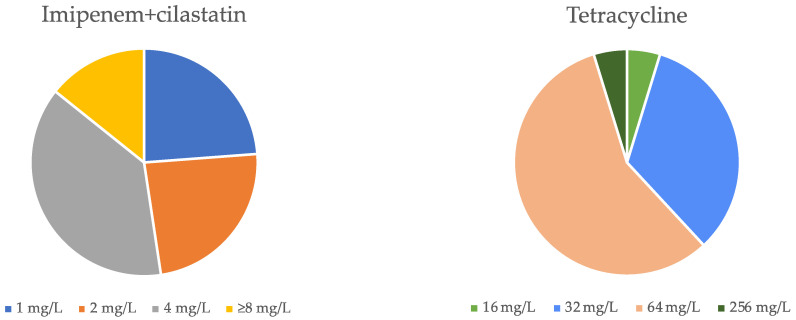
MIC values for imipenem + cilastatin and tetracycline antibiotics for the *P. aeruginosa* panel tested.

**Figure 3 microorganisms-11-02687-f003:**
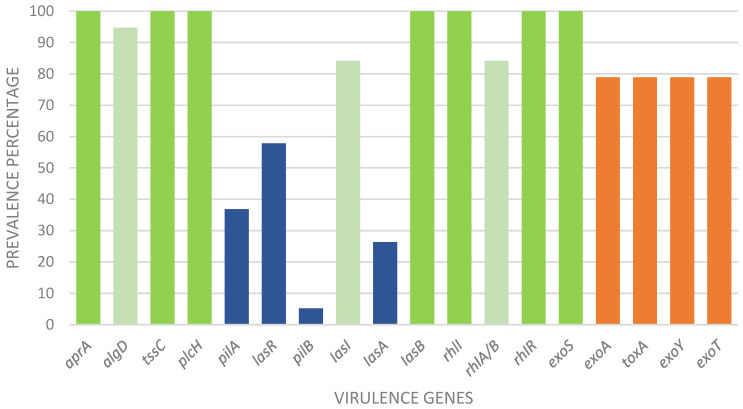
The percentage of each virulence gene found in septicemia-derived *P. aeruginosa* isolates from the Centro Médico de Trás-os-Montes e Alto Douro, Vila Real, Portugal, between September 2021 and June 2022. The dark green color represents percentages of 100%, the lighter green is used for percentages between 60% and 80%, orange for percentages between 60% and 80%, and blue for percentages below 60%.

**Table 1 microorganisms-11-02687-t001:** Synergistic effect of antibiotic combinations according to their concentration on biofilm formation ability of 6 *P. aeruginosa* isolates.

mg/L		TETRACYCLINE								
		1	2	4	8	16	32	64	128	256
IMI + CILAST	1									HHS12; HS17; HS18
	2							HS7; HS17	HS12; HS18	HS2
	4					HS7	HS17	HS20	HS2	
	8			HS7		HS17	HS18	HS12		
	16		HS7			HS18	HS20	HS2		
	32					HS20				
	64									

HS2, HS7, HS12, HS17, HS18, HS20: selected strains from the panel of 21 *P. Aeruginosa* septicemia isolates. Colors represent biofilm formation inhibition per sample; the green color is represented by more than one isolate.

**Table 2 microorganisms-11-02687-t002:** Characteristics and distribution of the genotypes detected among the examined *P. aeruginosa* strains.

Isolate	Aminoglycosides		Class D β-Lactamic	Class B MBLs						Class A Carbapenemases	Class A ESBLs	Other
	*aac*(3)-III	*aac*(3)-II	*bla_OXA-48_*	*bla_SPM_*	*bla_NDM_*	*bla_IMP_*	*bla_VIM_*	*bla_VIM2_*	*bla_PER_*	*bla_KPC_*	*bla_TEM_*	*oprD*
HS1	−	−	−	+	−	−	+	−	+	−	−	+
HS2	−	−	−	+	−	−	−	−	−	−	−	+
HS3	−	−	−	+	−	+	−	−	−	−	+	+
HS4	−	−	−	−	+	−	−	−	−	−	−	+
HS5	−	−	+	+	−	−	−	−	−	−	−	+
HS6	−	−	+	−	−	−	−	−	−	−	−	+
HS7	+	+	+	−	−	−	−	+	−	−	−	+
HS8	−	−	+	+	−	−	−	−	−	−	−	+
HS9	+	+	+	+	−	+	−	−	−	−	−	+
HS10	−	+	+	+	−	−	−	−	−	−	−	+
HS11	−	−	+	+	−	−	−	+	−	−	−	+
HS14	−	−	−	+	−	−	−	−	−	−	−	+
HS15	−	−	−	+	−	−	−	−	−	−	−	+
HS16	−	−	−	+	−	−	−	−	−	−	−	+
HS17	−	−	−	+	−	−	−	−	−	−	−	+
HS18	−	−	−	+	−	−	−	−	−	−	−	+
HS19	−	−	−	−	−	−	−	−	−	−	−	+
HS20	−	−	−	−	−	−	−	−	−	−	−	+
HS21	−	−	−	−	−	−	−	−	−	+	−	+

(+): Presence of a particular gene; (−): absence of a particular gene.

## Data Availability

Data are contained within the article.

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
