# Peer review of "Determination of Antimicrobial Resistance and the Impact of Imipenem + Cilastatin Synergy with Tetracycline in Pseudomonas aeruginosa Isolates from Sepsis"

_microorganisms, 2023, doi:10.3390/microorganisms11112687_

Round 1

Reviewer 1 Report

Comments and Suggestions for Authors

"The work has appeared very well-written and executed to me. I have only found a few details to correct."

·       Line 144: "Two of the columns of each plate" -> "Two columns of each plate"

·       Line 156: "presence/absence of biofilms depending on" -> "presence or absence of biofilms depending on"

·       Line 160: "nucleic acid concentrations were determine by measurements" -> "nucleic acid concentrations were determined by measurements"

  • Line 206: "in Egypt shows 51%" -> "in Egypt showed 51%"
  • Line 221: "Concerning ciprofloxacin, our study indicated a lower percentage of sensitive isolates," -> "Regarding ciprofloxacin, our study indicated a lower percentage of sensitive isolates,"
  • Line 249: "antibiotic removal. In simpler terms, this implies a potential role of efflux pumps in conferring resistance [23]." -> "antibiotic removal, implying a potential role of efflux pumps in conferring resistance [23]."
  • Line 251: "Ranji and Rahbar Takrami differ [24]." -> "Ranji and Rahbar Takrami differed in their results [24]."
  • Line 259: "leading to the death of the bacteria [8]." -> "leading to bacterial cell death [8]."
  • Line 280: "In this synergy test, 6 of the 21 isolates were chosen for this studied." -> "In this synergy test, 6 of the 21 isolates were selected for this study."
  • Line 286: "8 μg/mL of tetracycline and 64 μg/mL of imipenem+cilastatina." -> "8 μg/mL of tetracycline and 64 μg/mL of imipenem+cilastatin."
  • Line 298: "cell wall synthesis by β-lactams significantly enhances the absorp- 297 tion of other drugs [31]." -> "cell wall synthesis by β-lactams significantly enhances the absorption of other drugs [31]."
  • Line 318: "the optical densities (OD) of the negative control of each microplate were calculated and the calculations were performed using the following formula: ODc = mean OD of the negative control + (3 x SD of the negative control) [32]." -> "the optical densities (OD) of the negative control of each microplate were calculated using the following formula: ODc = mean OD of the negative control + (3 x SD of the negative control) [32]."
  • Line 322: "OD ≤ ODC (non-producer), ODC < OD ≤ 2ODC (weak), 2ODC < OD ≤ 4ODC (moderate), 4ODC > ODC (strong) [33]." -> "OD ≤ ODC (non-producer), ODC < OD ≤ 2ODC (weak), 2ODC < OD ≤ 4ODC (moderate), OD < 4ODC  (strong) [33]."
  • Line 340: "37°C was the temperature at which biofilms are formed with greater mechanical stability [37]." -> "37°C was the temperature at which biofilms were formed with greater mechanical stability [37]."
  • Line 348: "human temperature seems to be the favorable condition for this bacterium, which is why infections of this type are so problematic." -> "human body temperature seems to be the favorable condition for this bacterium, which is why infections of this type are so problematic."
  • Line 350: "infections of this type are so problematic." -> "infections caused by this bacterium are particularly challenging."
  • Line 358: "only 2 mg/L tetracycline but 16 mg/L imipenem+cilastatin." -> "only 2 mg/L of tetracycline but 16 mg/L of imipenem+cilastatin."
  • Line 364: "these inhibitory effects of imipenem+cilastatin and tetracycline combinations on bacterial biofilms formation are supported by other studies [38,39]." -> "these inhibitory effects of imipenem+cilastatin and tetracycline combinations on bacterial biofilm formation are supported by other studies [38,39]."
  • Line 378: "It should be noted that our results concerning the inhibition of biofilm production" -> "It is important to note that our results regarding the inhibition of biofilm production."
  • Line 382: "specific conditions of the bacterial infection in question and the safe and effective doses recommended for the clinical use of antibiotics." -> "specific conditions of the bacterial infection in question, and one must consider the safe and effective doses recommended for the clinical use of antibiotics."
  • Line 384: "This finding emphasized that the inhibition of bacterial growth does not necessarily guarantee the eradication of the biofilm." -> "This finding emphasizes that inhibiting bacterial growth does not necessarily guarantee the eradication of the biofilm."
  • Line 388: "The genotypic results for the rDNA 16S gene were positive for all isolates, allowing us to conclude that all of them were P. aeruginosa, as expected." -> "The genotypic results for the rDNA 16S gene were positive for all isolates, confirming their identification as P. aeruginosa, as expected."
  • Line 390: "Although an analysis was carried out on multiple genes, for the class of aminoglycosides, 15.79% of the isolates had the aac(3)-II gene and 10.53% the aac(3)-III gene (Table 3)." -> "Although an analysis was carried out on multiple genes, within the class of aminoglycosides, 15.79% of the isolates had the aac(3)-II gene, and 10.53% had the aac(3)-III gene (Table 3)."
  • Line 398: "bacteria or preventing their binding to bacterial ribosomes, where the synthesis of antibiotics occurs. proteins." -> "bacteria or preventing their binding to bacterial ribosomes, where the synthesis of proteins occurs.
  • Line 406: "with specific focus on the aac(3)-II and aac(3)-III genes, in high-risk international clones of P. aeruginosa high-risk clones are known for their ability to spread and cause difficult-to-treat infections, often associated with multi-drug resistance [41]." -> "with a specific focus on the aac(3)-II and aac(3)-III genes, in high-risk international clones of P. aeruginosa. High-risk clones are known for their ability to spread and cause difficult-to-treat infections, often associated with multi-drug resistance [41]."
  • Line 428: "For the identification of genes encoding Class D β-lactamic," -> "For the identification of genes encoding Class D β-lactamase enzymes,"
  • Line 446: "With regard to the detection of Class B MBLs, several genes were detected." -> "Regarding the detection of Class B MBLs, multiple genes were identified."
  • Line 546: "in 78.95% of the isolates." -> "in 78.95% of the isolates."
  • Line 556: "Interestingly, and accordingly to our results, the literature gene does not appear in a constant way and is related to the type of infection." -> "Interestingly, and according to our results, the presence of the exoU gene in the literature does not appear consistently and is related to the type of infection."

Please can you explain me this sentence (line 580) ]. Variation in lasA gene recognition may influence the specific populations and clinical settings studied.

Comments on the Quality of English Language

There are few issues to take into account.

Author Response

The authors thank all reviewers for their constructive comments, which allowed significant improvement of the manuscript. We proceeded with the revision of the manuscript in the light of the comments received and brief responses to the reviewers’ comments are included. All the modifications in the text are marked using “Track Changes” function.

Please see the attachment "Reviewer 1 comments and corrections" 

Reviewer 2 Report

Comments and Suggestions for Authors

microorganisms-2665869

Journal: Microorganisms

Manuscript ID: microorganisms-2665869

Title: Determination of antimicrobial resistance and the impact of imipenem+cilastatin synergy with tetracycline in Pseudomonas aeruginosa isolates from sepsis.

This work gives detail some information valuable insights for the ongoing battle against P aeruginosa infections.

Also, the present research is highlighting on the need for tailored antibiotic therapies and innovative approaches to combat biofilm-related resistance. However, the work is interesting; authors should done great efforts in improving their work to be suitable for publication.

I suggested major revision.

Here general comments for authors:

1.      Separate the results and discussion according to the journal structure.

2.      English editing is required.

3.      Font and style of the words should be in appropriate way.

4.      Figure 2 must be reconstructed.

5.      Line 304 Only one isolate formed strong biofilms. Add “has strong biofilms”.

6.      I suggest changing Table 1 to Figure, it will be more attractive.

7.      Conclusion: Should be summarized.  Also future directions such one sentence I suggest must be added.

Comments on the Quality of English Language

Extensive editing of English language required

Author Response

The authors thank all reviewers for their constructive comments, which allowed significant improvement of the manuscript. We proceeded with the revision of the manuscript in the light of the comments received and brief responses to the reviewers’ comments are included. All the modifications in the text are marked using “Track Changes” function.

Please see the attachment "Reviewer 2 comments and corrections"

Reviewer 3 Report

Comments and Suggestions for Authors

The Authors wrote that they “isolated strains using VITEK 2 Compact” what do you mean by that?  We use VITEK  to identify strains and estimate susceptibility to antibiotics. I’ve never heard about using it for isolation. Explain it.

In lines 106, 111, 112, 130, 141 the Authors should add the names of the companies

Fig. 1 in EUCAST there is no category intermediate, there is susceptible higher exposure. The Authors should change it. In EUCAST there are breakpoints for ceftazidim for P. aeruginosa, so the Authors should used them for all antimicrobials, not CLSI.

Line 192 and 205. The same citation, but in different way, line 192 Anjum and Mir, line 205 Anjum et al. Why?

Fig. 2 I don’ t understand why there’s a figure. Both categories had 0% and it’d have been written in the text. The figure is unnecessary.

Line 384 The abbreviation of MIC should be developed earlier in the text, in Material & Methods section

The names of the genes should be unified, e.g only blaSMP, blaNDM

 In the all text units should be the same. Sometimes the Authors used mg/L (e.g. lines 239, 243...), sometimes ug/mL (e.g. lines 283, 284 ...)

Author Response

The authors thank all reviewers for their constructive comments, which allowed significant improvement of the manuscript. We proceeded with the revision of the manuscript in the light of the comments received and brief responses to the reviewers’ comments are included. All the modifications in the text are marked using “Track Changes” function.

Please see the attachment "Reviewer 3 comments and corrections" 

Reviewer 4 Report

Comments and Suggestions for Authors

I think that the manuscript entitled “Determination of antimicrobial resistance and the impact of imipenem+cilastatin synergy with tetracycline in Pseudomonas aeruginosa isolates from sepsis” is in principle suited for a publication in Microorganisms, Section “Antimicrobial Agents and Resistance”. This study is significant for understanding and combatting Pseudomonas aeruginosa infections, particularly in septicemias. It explores antibiotic susceptibility, biofilm formation, aminoglycoside resistance, and genetic diversity in these bacteria. The findings emphasize the need for tailored antibiotic therapies and innovative approaches to tackle biofilm-related resistance, providing valuable insights for healthcare practitioners and researchers. However, I do have a few minor remarks and comments.

Comments:

For consistency, please use the same concentration units for determining MICs, e.g. µg/mL throughout. Both µg/ml and mg/l appear in the text.

Line 147. The term "oven" is not commonly used in laboratory contexts. You may want to mention an incubator. Please add information about the company that manufactured the device in which the incubation was performed.

Lines 155-156. In the phrase, "followed by 30% acetic acid," it's unclear whether the acetic acid was used for staining or for a different purpose.

Lines 163-164. The sentence, "To determine the nucleic acid concentrations were determine by measurements of the absorbance at 260 and 280 nm," is unclear and contains grammar errors.

Lines 165-166. The sentence, "The PCR concentration of each sample was 200 μg/ml," appears to be describing the PCR concentration, but it's not clear. It might be related to DNA concentration.

Line 171. The phrase, "class D β- lactamic (blaOXA, blaOXA- 48)," is confusing.

Line 179. The sentence, "T3SS effector proteinss (exoU, exoS, exoA, exoY, exoT)," contains a typo. It should be "proteins," not "proteinss."

Lines 179-180. The phrase, "rhl quorum sensing system (rhlR, rhlI, rhlA/B)," lacks clarity and should be rephrased, "rhl quorum sensing system genes (rhlR, rhlI, rhlA, rhlB)."

Lines 194-195. The sentence, "This intermediate status indicates that these bacterial strains exhibit an intermediate response," contains redundancy. "Intermediate" already indicates a middle ground. Suggestion: "This status indicates that these bacterial strains exhibit an intermediate response."

Lines 210-212. In the sentence, "However, in the specific context of the study at hand, the range of efficacy values for imipenem was notably wider than ever documented before," it's not clear what "ever documented before" refers to. It might be helpful to specify the timeframe or context you are comparing to. Suggestion: "However, in the specific context of this study, the range of efficacy values for imipenem was notably wider than previously documented in similar studies."

Lines 243-244. In the sentence, "The consistency of results in the face of a resistant phenotype (> 8 mg/L) is probably due to the high efficiency of P. aeruginosa's mechanisms of action against this antibiotic," you can rephrase it for clarity. "The consistency of resistant phenotypes (MIC > 8 mg/L) suggests that P. aeruginosa employs highly effective mechanisms against this antibiotic."

Lines 273-274. In the sentence, "Carbapenems are able to resist the hydrolytic action of the β-lactamase enzyme, which makes them more effective in fighting bacterial infections," it's important to clarify that carbapenems are not resisting the enzyme but rather are resistant to its action.

Lines 364-293. I think the imipenem+cilastatin combination was not used in the cited papers #38, 39. Please check and either provide relevant links or rephrase the sentence.

Lines 434-437. Perhaps this refers to the work of the authors Tarafdar, F, et al. and not Dubois, V. et al. Please check this. Also, please check that the numbering of the cited works is not incorrect.

Line 463. “A study guided by Sharma et al.” I didn’t find the corresponding authors in the References section.

Please check the accuracy of the publication data of articles in the References section.

Author Response

The authors thank all reviewers for their constructive comments, which allowed significant improvement of the manuscript. We proceeded with the revision of the manuscript in the light of the comments received and brief responses to the reviewers’ comments are included. All the modifications in the text are marked using “Track Changes” function.

Please see the attachment "Reviewer 4 comments and corrections" 

Round 2

Reviewer 2 Report

Comments and Suggestions for Authors

authors addressed all the comments. 

Comments on the Quality of English Language

NONE